# Sugar Transporter, CmSWEET17, Promotes Bud Outgrowth in *Chrysanthemum Morifolium*

**DOI:** 10.3390/genes11010026

**Published:** 2019-12-24

**Authors:** Weixin Liu, Bo Peng, Aiping Song, Jiafu Jiang, Fadi Chen

**Affiliations:** State Key Laboratory of Crop Genetics and Germplasm Enhancement, Key Laboratory of Landscaping, Ministry of Agriculture and Rural Affairs, College of Horticulture, Nanjing Agricultural University, Nanjing 210095, China; lwx060624@163.com (W.L.); pengbo96651@163.com (B.P.); aiping_song@njau.edu.cn (A.S.); jiangjiafu@njau.edu.cn (J.J.)

**Keywords:** *CmSWEET17*, sucrose, bud outgrowth, *Chrysanthemum morifolium*, auxin

## Abstract

We previously demonstrated that 20 mM sucrose promotes the upper axillary bud outgrowth in two-node stems of *Chrysanthemum morifolium*. In this study, we aimed to screen for potential genes involved in this process. Quantitative reverse transcription (qRT)-PCR analysis of sugar-related genes in the upper axillary bud of plants treated with 20 mM sucrose revealed the specific expression of the gene *CmSWEET17*. Expression of this gene was increased in the bud, as well as the leaves of *C. morifolium,* following exogenous sucrose treatment. *CmSWEET17* was isolated from *C. morifolium* and a subcellular localization assay confirmed that the protein product was localized in the cell membrane. Overexpression of *CmSWEET17* promoted upper axillary bud growth in the two-node stems treatment as compared with the wild-type. In addition, the expression of auxin transporter genes *CmAUX1*, *CmLAX2, CmPIN1, CmPIN2,* and *CmPIN4* was upregulated in the upper axillary bud of *CmSWEET17* overexpression lines, while indole-3-acetic acid content decreased. The results suggest that *CmSWEET17* could be involved in the process of sucrose-induced axillary bud outgrowth in *C. morifolium*, possibly via the auxin transport pathway.

## 1. Introduction

The chrysanthemum, one of the most well-known flowers in the world, originated in China and possesses great ornamental and economic value. However, the production of single-headed cut chrysanthemum flowers requires a large amount of manpower and material resources to manually remove the axillary buds. This labor-intensive process limits the development of the ornamental chrysanthemum industry to a certain extent [1]. Currently, reports on the shoot branching of chrysanthemum focus mainly on the effects of hormones, including strigolactone [2], auxin [3], and cytokinin [4], as well as other factors such as temperature [5] and transcription factor *BRC1* [6]. However, few studies have reported on the effect of sugar on the regulation of the shoot branching of chrysanthemum [7].

Sugar, as both an energy source and a signaling molecule, plays a crucial role in processes of plant growth and development, including flowering [8,9], meristematic proliferation [10], and anthocyanin biosynthesis [11]. Studies have shown that sugar also affects shoot branching and bud outgrowth [12,13]. In the rose plant, exogenous sugars (sucrose, palatinose, and psicose) promote axillary bud growth in vitro [14]. Furthermore, budbreak in the walnut plant is associated with the sucrose absorption [15]. Additionally, gentiobiose, a rare disaccharide, acts as a signal to promote bud growth in *Gentiana*, and exogenous gentiobiose facilitates bud germination [16].

Since 1934, when Thimann and Skoog demonstrated that apical dominance was associated with auxin [17], this hormone has been regarded as the key signal controlling bud outgrowth and apical dominance [18]. The growth of axillary buds is accompanied by auxin export from the buds and an increase in the expression of the auxin transport gene *PINs* [4,19]. In recent years, studies have shown that the initial signaling molecule regulating plant apical dominance could be sugar rather than auxin [20,21]; and that auxin acts on subsequent continuous growth stages [22]. However, the mechanism by which sugar regulates axillary bud outgrowth remains unclear.

In plants, sugar is produced mainly by photosynthesis in leaves, from where it is then transported to other parts of the plant to serve its function. This process requires specific sugar transporting proteins, such as sucrose transporters SUC/SUT and bidirectional sugar transporters SWEET [23]. Sugar transporters are indispensable for sugar function. Sugar transport protein 1 (STP1) regulates shoot branching by controlling the expression of genes involved in hormone biosynthesis and signaling in *Arabidopsis thaliana* [24].

SWEET proteins belong to the MtN3/saliva family, with the majority being composed of seven predicted α-helical transmembrane (TM) domains, and a few of only three TM domains [25]. The N- and C-terminals are located in the inner and outer cytoplasm, respectively [26]. In *A. thaliana* and rice, SWEETs participate in phloem transport of sucrose [27], nectar secretion [28], and regulating gibberellin-mediated physiological processes [29]. There are 17 SWEET members in *A. thaliana* [25,30]. AtSWEET17, which belongs to the fourth subfamily [31], participates in the regulation of fructose accumulation in leaves, as demonstrated by the decrease in fructose content of leaves in *sweet17-1* and *sweet17-2* mutants [32]. *SWEET17* expression can be induced by fructose and darkness; the fructose content, in the leaves of *SWEET17* overexpression lines, has been shown to decrease sharply under cold stress [33]. The expression of DsSWEET17 in *Dianthus spiculifolius* is affected by exogenous fructose, glucose, salt stress, and osmotic stress and following its transformation into *A. thaliana*, root growth was promoted [31]. Transgenic lines of PtSWEET17a overexpression in poplar result in a growth promoting phenotype, which is manifested as a thicker stem diameter and increased plant height [34]. However, to the best of our knowledge, the involvement of *SWEET17* in shoot branching has not been reported to date.

A previous study found that treatment with 20 mM sucrose could promote axillary bud outgrowth in *C. morifolium* as compared with the control (Appendix A) [35]. In this study, a quantitative analysis was performed to screen for key genes involved in this process, revealing differential expression of the gene *CmSWEET17*. To further investigate the role of this gene, transgenic engineering, gene expression quantification, and auxin content determination were carried out.

## 2. Materials and Methods

### 2.1. Plants and Growth Conditions

Plantlets of chrysanthemum (*Chrysanthemum morifolium* ‘Jinba’) were propagated under sterile conditions with MS medium, followed by growth in a tissue culture room at 23 °C with a photoperiod of 16 h light and 8 h dark and a light intensity of 100 to approximately 120 µmol photons·m^−2^·s^−1^. 

### 2.2. Two-Node Stem Treatments

The two-node stem was established in order to study the development of axillary buds (Appendix A). When a seedling grew to ~10 cm, the stem was cut into a two-node stem containing two axillary buds. The two-node stems were treated with 0 or 20 mM sucrose, as described previously [35]. The treatments consisted of MS medium with 20 mM mannitol (control) or 20 mM sucrose. The treatment groups were grown under darkness to avoid photosynthesis-induced sucrose production. After 10 days of treatment, the phenotype of axillary bud length was changed and the axillary buds from each group were collected for subsequent RNA extraction. Due to the small size of the axillary buds, the sampling was repeated 4 to approximately 5 times, and each sample was taken at the same time of day.

### 2.3. Quantitative Reverse Transcription PCR (qRT-PCR) Analysis

Total RNA was extracted from the axillary buds using Trizol reagent according to the manufacturer’s protocol (Takara, Shiga, Japan). First-strand cDNA was synthesized using a PrimeScript™ RT Reagent kit (Takara). The qPCR reactions were run using the Mastercycler^®^ EP Realplex Real-Time PCR system (Eppendorf, Hamburg, Germany), as described previously [36]. Using the existing transcriptome library (http://www.ncbi.nlm.nih.gov/bioproject/PRJNA329030), we screened sugar-related genes, including signal key signaling genes hexokinase (*HXK)*, sucrose non-fermenting-related kinase 1 (*SnRK1*), and target of rapamycin (*TOR*), as well as sugar transporter *SUC* and *SWEET* genes. Gene-specific primers were designed using Primer Premier 5.0 (Premier Biosoft International, Palo Alto, CA, USA), and the sequences are listed in Appendix A. The chrysanthemum *CmEF1α* gene (GenBank acc. no., AB679278.1) was used as an internal control.

### 2.4. Exogenous Sucrose Treatment

When the *C. morifolium* seedlings grew to 15 cm and produced 10 true leaves, they were treated with 50 mM sucrose, as previously described [9]. Following treatment, the second top leaf (fully unfolded) and axillary bud were collected at 0, 0.5, 1, 3, 6, and 24 h. The leaves and axillary buds were snap frozen in liquid nitrogen and stored at −80 °C, for subsequent RNA extraction and quantitative analysis of *CmSWEET17*.

### 2.5. Isolation of CmSWEET17 and Sequence Analysis

The sequence of the sugar transporter gene *CmSWEET17* was obtained using ORFfinder (https://www.ncbi.nlm.nih.gov/orffinder/), and the full length was cloned using gene-specific primers (Appendix A). The PCR reaction was composed of 10 μL 5x Phusion HF Buffer, 2 μL each primer (10 μM), 1 μL dNTP (10 μM), 0.5 μL dimethyl sulfoxide (DMSO), 34 μL H_2_O, 0.5 μL Phusion High-Fidelity DNA Polymerases (Thermo Fisher Scientific, Inc., Waltham, MA, USA) and 0.5 μL of cDNA. The PCR cycling regime comprised an initial denaturation step at 98 °C for 30 s, followed by 40 cycles at 98 °C/10 s, 55 °C/30 s, and 72 °C/30 s, with a final extension step at 72 °C for 8 min. The resulting amplicon (876 bp in length) was introduced into the pMD19-T vector (Takara) for sequencing. After successfully obtaining the *CmSWEET17* sequence, prediction of the transmembrane domains of was performed using TMHMM (http://www.cbs.dtu.dk/services/TMHMM/). Amino acid sequence alignment and construction of a phylogenetic tree was carried out alongside AtSWEET family members from A. thaliana, using molecular evolutionary genetics analysis software DNAMAN version 6 (Lynnon Biosoft, San Ramon, CA, USA).

### 2.6. Subcellular Localization of the CmSWEET17 Protein

The open reading frame of the *CmSWEET17* gene was amplified using forward and reverse primers incorporating *Sal*I and *Not*I restriction sites, respectively (*CmSWEET17*-1A-F and *CmSWEET17*-1A-R, Appendix A). The PCR reagents and conditions were the same as Section 2.5. Restriction enzymes, *Sal*I and *Not*I, were used to digest the resulting PCR products and the pENTR™ 1A Dual Selection entry vector (Thermo Fisher Scientific, Inc.), and the double enzyme products were recovered by DNA Gel Extraction Kit (Axygen) and ligated with solution I (Takara). Subsequently, the resulting vector (pENTR1A-*CmSWEET17*) was used to construct a green fluorescent protein (GFP) fusion vector pMDC43-*CmSWEET17* by means of the Gateway^®^ LR Clonase^®^ II enzyme (Thermo Fisher Scientific, Inc.). The construct was transiently introduced into onion epidermal cells using a PDS-1000/He helium-driven particle accelerator (Bio-Rad Laboratories, Inc., California, USA), which vacuum level, helium pressure, and target interval were 26 to 28 inches Hg (mercury), 1300 psi (pounds per square inch), and 6 cm, respectively. And the cells were then incubated at 25 °C for 16 h in the dark. The GFP fluorescence was monitored by laser scanning confocal microscopy (LSM780; Zeiss, Oberkochen, Germany) at 488 nm. To induce plasmolysis, a few drops of 20% mannitol solution were added to the onion epidermis cells, which were then incubated at room temperature for ~20 min under dark conditions.

### 2.7. Chrysanthemum Transformation

The pMDC43-*CmSWEET17* plasmid was used as an overexpression vector in *Agrobacterium tumefaciens* strain EHA105 using the freeze-thaw method, and subsequently into *C. morifolium*, as described previously [37]. Hygromycin-resistant plants were obtained, and their DNA was extracted, as described previously [35], and verified by PCR analysis (primers pMDC43-F and *CmSWEET17*-R, Appendix A). After the DNA verification, the potential overexpression lines (OX) of *CmSWEET17* were validated using qRT-PCR (primers *CmSWEET17*-Q-F/R, Appendix A). The overexpression lines were treated as per the two-node stem treatments described in Section 2.2, and the bud length was measured under a stereomicroscope, with buds from 10 stem segments measured for each treatment.

### 2.8. Indole-3-Acetic Acid (IAA) Content Measurement

The buds from the two-node stem treatment were harvested, flash frozen in liquid nitrogen, and stored at −80 °C (sampling as described in Section 2.2). Auxin (IAA) was extracted and measured using ultraperformance liquid chromatography-tandem mass spectrometry (UPLC-MS/MS), as described previously [38].

### 2.9. Statistical Analysis

Statistical analyses were carried out using Excel 2017 (Microsoft Corporation), and data are expressed as the mean ± SE of at least three separate experiments and *p* ≤ 0.05 was considered to indicate a statistically significant difference. The images were processed using Adobe Photoshop software. 

## 3. Results

### 3.1. Expression of CmSWEET17 in Response to Sucrose

To screen for genes that potentially serve a role in sucrose-induced promotion of axillary bud outgrowth in the two-node stem system (Appendix A), a qRT-PCR analysis of sugar-related genes was performed. The only gene detected to be specifically expressed in the upper bud of plants treated with 20 mM sucrose was *CmSWEET17* (Figure 1). This gene could, therefore, be involved in the process of sucrose-promoted outgrowth of the upper bud.

To verify whether sucrose affects the expression of *CmSWEET17*, we sprayed *C. morifolium* seedlings with 50 mM sucrose solution and analyzed the gene expression profile in the leaves using qRT-PCR. The data showed that the treatment resulted in a rapid increase in *CmSWEET17* expression, peaking at 30 min, when it was 2.52× higher than the baseline level (Figure 2A). The expression level then decreased gradually over time, returning to baseline at 24 h (Figure 2A). In the axillary buds, the expression of *CmSWEET17* increased gradually from 0 to 24 h, at 24 h, when it was 2.08× higher than the baseline level (Figure 2B). The results demonstrate that exogenous sucrose treatment can lead to an increase in the expression of *CmSWEET17* in the leaves and lateral buds of *C. morifolium*.

### 3.2. CmSWEET17 Sequence Analysis

To clone the *CmSWEET17* gene, we designed the specific primers (*CmSWEET17*-F and *CmSWEET17*-R, Appendix A). The open reading frame (ORF) of *CmSWEET17* was found to be 876 bp, which encoded 291 amino acids. Phylogenetic analysis revealed that CmSWEET17 was most closely related to AtSWEET17, belonging to clade IV of the Arabidopsis AtSWEET family (Figure 3A). Using the TMHMM algorithm, CmSWEET17 was predicted to have six transmembrane domains (Figure 3B), SWEET family genes in other species generally contain seven transmembrane domains. CmSWEET17 lacks a second transmembrane domain (TM2) as compared with AtSWEET17 in Arabidopsis thaliana (Figure 3C).

### 3.3. Subcellular Localization of CmSWEET17

In order to determine the localization of the CmSWEET17 protein in living cells, we constructed the pMDC43-*CmSWEET17* vector, delivered it into onion inner epidermis cells, and observed GFP signals. These signals were observed only in the cell membrane (Figure 4). To further verify whether CmSWEET17 was located in the cell membrane, 20% mannitol solution was used for plasmolysis. With the separation of the plasma membrane from the cell wall, the GFP signals also moved along with the cell membrane (Figure 4). These data indicate that CmSWEET17 is located in the cell membrane.

### 3.4. Effect of CmSWEET17 Overexpression on Bud Outgrowth

To verify the function of *CmSWEET17*, the overexpression vector pMDC43-*CmSWEET17* was constructed and transformed into *C. morifolium* via the *Agrobacterium*-mediated leaf disc method, and overexpression lines were obtained. PCR analysis of the genomic DNA identified the OX-39 and OX-40 lines of *CmSWEET17* (Appendix A). Subsequently, quantitative analysis revealed that *CmSWEET17* expression in OX-39 and OX-40 was increased by 4.01 and 3.01 times, respectively, as compared with the control (Figure 5A).

In order to test whether *CmSWEET17* regulates the outgrowth of the axillary bud, the two-node stem treatment was applied to the OX-39, OX-40, and wild type (WT) lines. After 10 days, the length of upper and lower buds of the WT plant were 5.67 and 1.04 mm, respectively, whereas the upper buds of OX-39 and OX-40 were 8.11 and 7.06 mm, respectively, and their lower buds were 2.01 and 1.67 mm, respectively (Figure 5B,C). These results indicate that *CmSWEET17* served a role in the promotion of the axillary bud outgrowth in *C. morifolium*.

### 3.5. Effect of CmSWEET17 Overexpression on Auxin Export from Buds

In order to explore the possible pathway by which CmSWEET17 promotes axillary bud outgrowth, we performed quantitative analyses on the axillary buds from overexpression lines following two-node stem treatment. The results showed that the expression levels of auxin transporter genes *CmAUX1*, *CmLAX2*, *CmPIN1*, *CmPIN2,* and *CmPIN4* in the upper buds of the OX-39 and OX-40 lines were increased as compared with those in the WT plants (Figure 6A). Further determination of IAA content revealed that the levels in the WT, OX-39, and OX-40 lines were 8.16, 7.49, and 6.02 pg.mg^−1^, respectively, indicating a significant decrease of auxin content in the overexpression lines (Figure 6B). These data imply that *CmSWEET17* overexpression may induce auxin export from buds by upregulating the expression of auxin transporter protein, thus, promoting axillary bud outgrowth.

## 4. Discussion

Sucrose is a critical energy and signaling molecule in plants, playing an important role in a number of processes related to growth and development [39]. Previous studies of our group have demonstrated that 20 mM sucrose promotes the upper bud outgrowth in the two-node stem system in *C. morifolium* [35]; therefore, we speculated that sugar-related genes are involved in this process. Using the existing transcriptome library, the expression levels of sugar-related genes *HXK*, *SnRK1*, *TOR*, *SUCs,* and *SWEETs* were measured, revealing that *CmSWEET17* was highly expressed in the growing apical buds (Figure 1). Furthermore, exogenous sucrose treatment led to an increase in the expression of *CmSWEET17* in both the leaves and axillary buds of *C. morifolium* (Figure 2). In *A. thaliana*, SWEET17, a fructose transport protein, was found to respond to the absence of light and fructose [33]. Likewise, in *C. morifolium*, the expression of *CmSWEET17* responded to sucrose. This suggests that CmSWEET17 could be involved in the regulation of sucrose-induced promotion of axillary bud outgrowth in *C. morifolium*, which is positively correlated with the axillary bud growth.

Two types of mechanisms are known to be involved in the unloading of sucrose in sink tissues, the symplasmic and apoplasmic pathways. In the symplasmic pathway, sucrose is transported directly to the sink cells through plasmodesmata, without requiring transporters and energy consumption. Conversely, in the apoplasmic pathway, sucrose is first degraded into monosaccharides (fructose and glucose), then transported to parenchyma cells by monosaccharide transport proteins, and finally into the sink cells through plasmodesma. CmSWEET17, a fructose transport protein, therefore, could be involved in the apoplasmic pathway of sucrose unloading in sink tissues, a hypothesis which requires further study.

Bioinformatics analysis of the CmSWEET17 protein revealed that it contains six TM domains (Figure 3B), while the equivalent SWEET17 protein in other species contains seven [31,33]. Moreover, studies in other species have found that SWEET17 is localized mainly on the vacuole membrane [31,32], whereas the results of this study revealed that, in *C. morifolium*, CmSWEET17 is localized on the cell membrane (Figure 4). Whether this difference in localization is related to the absence of a TM domain in the structure of CmSWEET17 remains unknown.

Transgenic experiments were conducted to verify whether *CmSWEET17* is involved in the regulation of axillary bud outgrowth. After obtaining CmSWEET17 overexpression lines (OX-39 and OX-40), the two-node stem system was treated with 20 mM sucrose. The results showed that axillary bud outgrowth was promoted in the OX-39 and OX-40 lines (Figure 5). In the rose, the growth of axillary buds was found to be positively correlated with sugar metabolism [40], and the expression of sucrose transporter RhSUC2 was shown to be increased during the progress of axillary bud outgrowth [41]. In *A. thaliana*, the regulation of shoot branching by sugar transporter protein 1 (AtSTP1) depended on extracellular sugar content [24]. A quantitative analysis in this study showed that the expression of auxin transport genes *CmAUX1*, *CmLAX2*, *CmPIN1*, *CmPIN2,* and *CmPIN4* was increased in the axillary buds of the OX-39 and OX-40 lines (Figure 6A), suggesting in increase in auxin transport. Further determination of auxin content revealed that the levels decreased in the *CmSWEET17* overexpression lines (Figure 6B). Export of auxin from axillary buds is key to their growth [42] and requires the participation of auxin transport proteins [18]. During the growth of plant axillary buds, expression of auxin transport genes, *PIN1* and *AUX1,* was found to be increased [43,44]. Sugar has been regarded as the initial regulator to promote axillary bud outgrowth in recent years, and auxin is known to be involved in subsequent continuous growth stages [20,22]. In the rose and pea, sucrose could be the upstream signal of the key hormonal mechanisms controlling bud outgrowth. Sucrose has been shown to induce the expression of *RhPIN1*, leading to an increase in auxin export flow, promoting the growth of axillary buds [21]. The results of this study showed that, in *C. morifolium*, CmSWEET17 could promote the axillary bud outgrowth by the regulating auxin transport pathway. However, the exact mechanism by which CmSWEET17 regulates auxin transport to promote the axillary bud outgrowth requires further investigation.

## 5. Conclusions

In this study, *CmSWEET17* was found to be upregulated in the upper axillary buds of the two-node stem system following treatment with 20 mM sucrose. The data provide a preliminary confirmation of a novel function of *CmSWEET17* in promoting axillary bud outgrowth. Furthermore, this function could occur via the auxin transporter pathway, increasing the expression of auxin transporters, as well as auxin export from the axillary bud.

## Figures and Tables

**Figure 1 genes-11-00026-f001:**
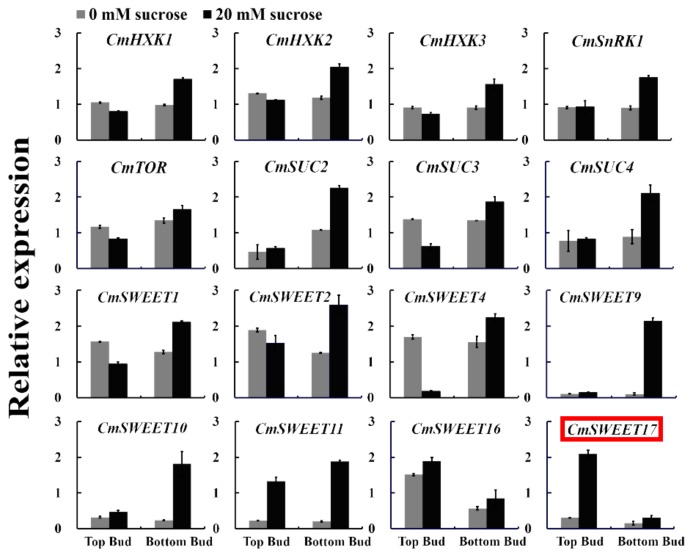
Relative expression of sugar-related genes in the axillary bud following the two-node stems treatment.

**Figure 2 genes-11-00026-f002:**
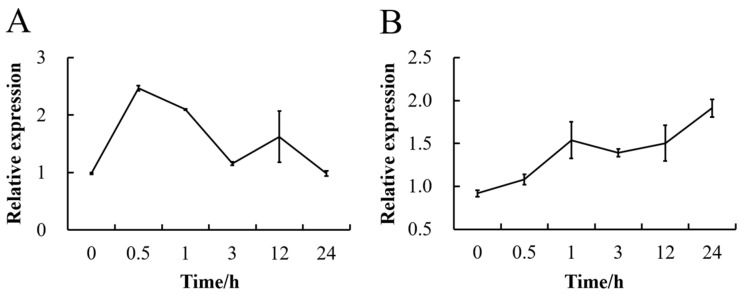
Change in expression level of the *CmSWEET17* gene in response to exogenous sucrose treatment (**A**) in the leaf, and (**B**) in the axillary bud.

**Figure 3 genes-11-00026-f003:**
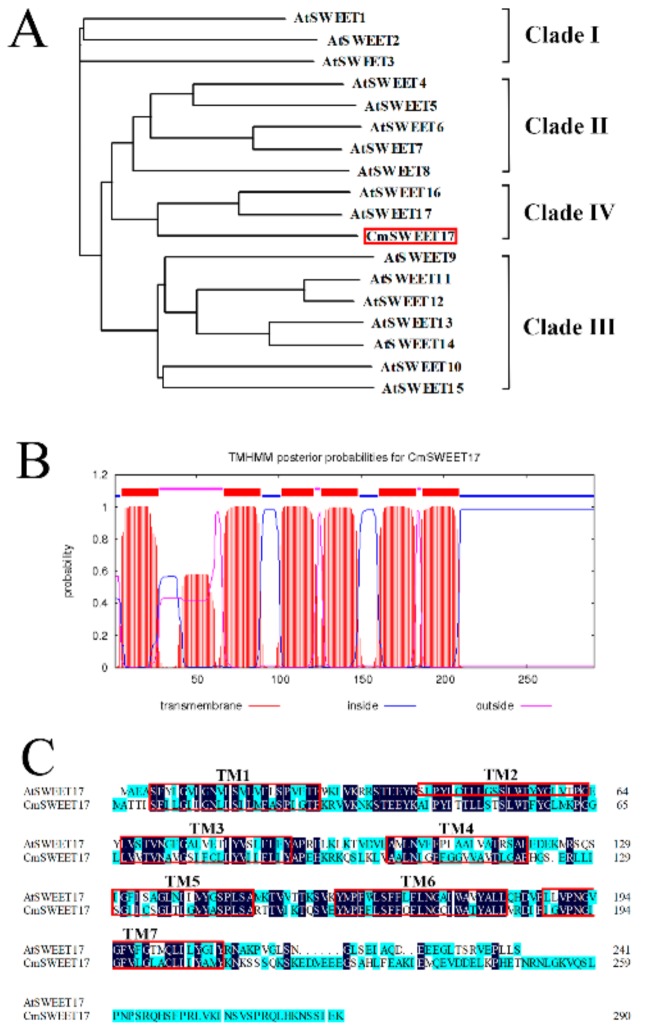
CmSWEET17 sequence analysis. (**A**) Phylogenetic tree of sequences including CmSWEET17 and AtSWEET family members from Arabidopsis thaliana. (**B**) Putative transmembrane TM domains within the CmSWEET17 sequence. (**C**) Amino acid sequence alignment of CmSWEET17 against AtSWEET17. Red boxes indicate TM domains and TM, transmembrane.

**Figure 4 genes-11-00026-f004:**
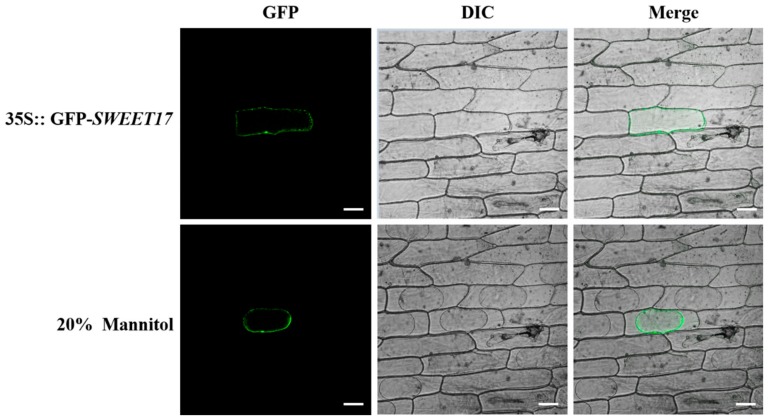
Subcellular localization of CmSWEET17 protein in onion epidermis cells. GFP, green fluorescence channel; DIC, bright light channel; and MERGE, overlay plots. Scale bars, 50 μm.

**Figure 5 genes-11-00026-f005:**
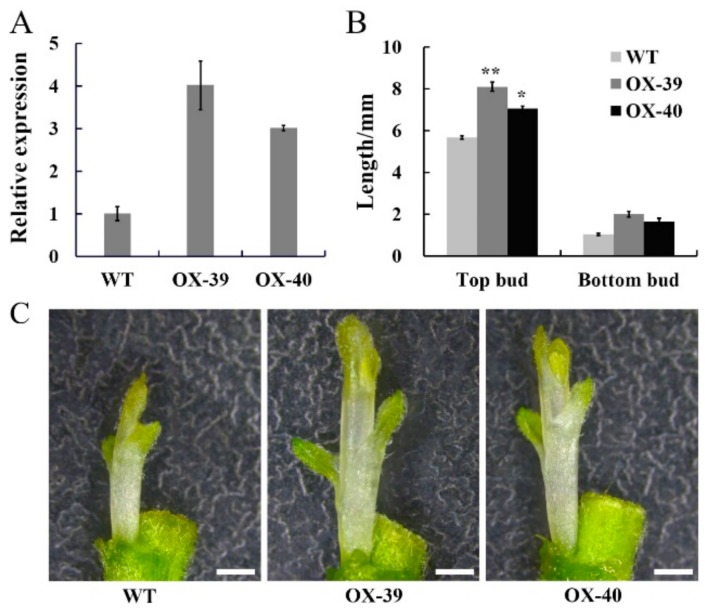
*CmSWEET17* overexpression lines promote bud growth in *Chrysanthemum morifolium*. (**A**) Relative CmSWEET17 expression in WT (wild-type) and overexpression lines OX-39 and OX-40 and (**B**) the length of the bud following treatment of two-node stem. Data are represented as the mean ± SE of *n* = 10. (**C**) Phenotype of axillary bud growth in WT, OX-39, and OX-40. Scale bars, 1 mm. * *p* ≤ 0.05 and ** *p* ≤ 0.01 vs. WT.

**Figure 6 genes-11-00026-f006:**
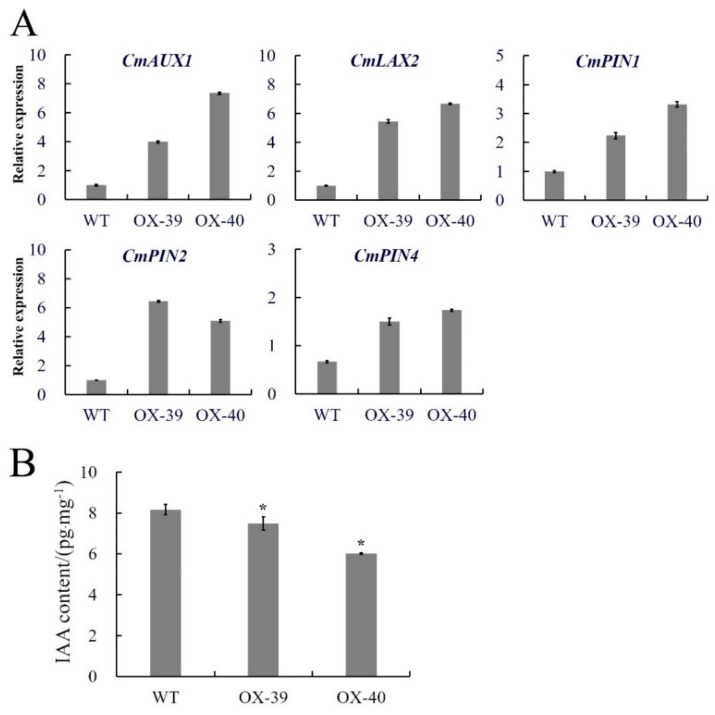
Differences in transgenic *CmSWEET17* overexpression lines. (**A**) Relative gene expression and (**B**) IAA content analysis of WT and overexpression lines (OX-39 and OX-40). * *p* ≤ 0.05 vs. WT. WT, wild-type and IAA, indole-3-acetic acid.

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
