# Peer review of "Sugar Transporter, CmSWEET17, Promotes Bud Outgrowth in Chrysanthemum Morifolium"

_genes, 2019, doi:10.3390/genes11010026_

Round 1

Reviewer 1 Report

This study provides an analysis of the genes involved in the process of sucrose induced upper axillary bud outgrowth. The study was performed well and appears to show a link between sucrose transport via CmSWEET17 and increased auxin transport, resulting in axillary bud growth. The main problems with the paper are the English, with various grammatical errors and awkward sentences, a need for greater detail in the methodology section, and further information required in the introduction and discussion sections. I suggest adding the information detailed below and having the English edited by a professional before acceptance of this manuscript.

Introduction: The introduction needs to be reordered so as to state the problem that the paper is addressing at the beginning. This paper was written to help address issues with the Chrysanthemum industry and this needs to be discussed first. There also needs to be more information related to the role of auxin in plant growth, as the regulation of auxin transporters are key to this paper. A small section detailing how sugar sensing can lead to altered gene expression would also be useful. Additionally, the final paragraph of the introduction needs to be strengthened. It is not entirely clear from this paragraph what the author's objectives are.

Materials and Methods: Overall there are some descriptions of the methodology that are left out. Noticeably, the statistical analysis performed is missing. For example, how did you determine that the effect of overexpression of CmSWEET17 on axillary bud growth was statistically different to the wild type? Any other statistical analysis used in this study should also be mentioned. Regarding specific sections:

Section 2.3: How did you apply the 50mM sucrose treatment to the seedlings?
Section 2.5: This needs to be expanded to include more information. What was your procedure for cloning the gene and how did you sequence it? It should also be mentioned that the phylogenetic tree was constructed using AtSWEET family members from A. thaliana.
Section 2.6: This section is also missing some information. You should detail your method for inserting the ORF into the vector pENTR1A and how you undertook the LR reaction. It would also be good to mention that the vector pMDC43 carries the GFP gene rather than just listing GFP in the name of the transgene.
Section 2.7: How many overexpression lines were obtained and screened? You also lack a description of what was performed using these overexpression lines.

Results:
You mention in section 3.1 the type of genes screened in this study. This should be mentioned in the methods section.

Lines 162 – 163: This information should be included in your methods section. You do not mention in your methods that you imaged before and after adding the mannitol solution. Also, “mannose” should be changed to “mannitol”.

Discussion: It is not clear how the differences in transmembrane domains is important to this study. The discussion also lacks any perspective on how increased CmSWEET17 expression may affect the expression of auxin transporter genes. The author should give forth a reason for this. It seems possible that increased sucrose causes higher SWEET expression in order to store the extra sucrose. A higher presence of sucrose in the cell may then act as a signal to induce expression of auxin transport genes.

Conclusion: Your conclusion lacks any mention that CmSWEET17 expression appears to contribute to the expression of auxin transport genes, which in turn leads to altered transport of auxin and ultimately altered axillary bud growth.

A few typos:

Line 186: Typo in “Phenotype”.
Line 223: Typo in “SWEEET17”
Lines 373 – 375: Should be mM, not Mm. “Scale bars indicate 1 mm” is repeated.

Author Response

Response to Reviewer 1 Comments

Point 1: This study provides an analysis of the genes involved in the process of sucrose induced upper axillary bud outgrowth. The study was performed well and appears to show a link between sucrose transport via CmSWEET17 and increased auxin transport, resulting in axillary bud growth. The main problems with the paper are the English, with various grammatical errors and awkward sentences, a need for greater detail in the methodology section, and further information required in the introduction and discussion sections. I suggest adding the information detailed below and having the English edited by a professional before acceptance of this manuscript.

Response 1: Thank you very much for your advice of this manuscript. We have finished the revision based on the comments and have asked native-speaker English editing to polish the revision.

Point 2: Introduction: The introduction needs to be reordered so as to state the problem that the paper is addressing at the beginning. This paper was written to help address issues with the Chrysanthemum industry and this needs to be discussed first. There also needs to be more information related to the role of auxin in plant growth, as the regulation of auxin transporters are key to this paper. A small section detailing how sugar sensing can lead to altered gene expression would also be useful. Additionally, the final paragraph of the introduction needs to be strengthened. It is not entirely clear from this paragraph what the author's objectives are.

Response 2: We have put the paragraph of ‘Chrysanthemum industry and this needs’ at the top of introduction.

In the paragraph 3 of the introduction, we add the description of role of auxin in the plant shoot branching.

In the paragraph 4 of the introduction, we add the reference [24]: “sugar Transport Protein 1 (STP1) regulates shoot branching by controlling the expression of genes involved in hormone biosynthesis and signaling in Arabidopsis thaliana”.

In the final paragraph of the introduction, we have revised and highlighted the research focus.

Point 3: Materials and Methods: Overall there are some descriptions of the methodology that are left out. Noticeably, the statistical analysis performed is missing. For example, how did you determine that the effect of overexpression of CmSWEET17 on axillary bud growth was statistically different to the wild type? Any other statistical analysis used in this study should also be mentioned. Regarding specific sections:

Section 2.3: How did you apply the 50mM sucrose treatment to the seedlings?
Section 2.5: This needs to be expanded to include more information. What was your procedure for cloning the gene and how did you sequence it? It should also be mentioned that the phylogenetic tree was constructed using AtSWEET family members from A. thaliana.
Section 2.6: This section is also missing some information. You should detail your method for inserting the ORF into the vector pENTR1A and how you undertook the LR reaction. It would also be good to mention that the vector pMDC43 carries the GFP gene rather than just listing GFP in the name of the transgene.
Section 2.7: How many overexpression lines were obtained and screened? You also lack a description of what was performed using these overexpression lines.

Response 3: We have revised and added greater detail in the “Materials and Methods”. First, in the “section 2.9”, we add the method of statistical analysis.

Second, “Exogenous Sucrose Treatment” exchange orders with “Quantitative reverse transcription PCR (qRT-PCR) analysis”, exogenous 50 mM sucrose treatment is based on reference [9].

In the section 2.5, we add more information of isolation of CmSWEET17 and Sequence Analysis, including the PCR reaction and procedure and the soft of sequence analysis.

In the section 2.6, we add more information of the process of the vector pMDC43 construction and subcellular localization.

In the section 2.7, we have written the moore description of validation of overexpression lines. At first, we obtain about 60 lines. After PCR analysis, only 2 lines are verified successfully.

Point 4: Results:
You mention in section 3.1 the type of genes screened in this study. This should be mentioned in the methods section.

Lines 162 – 163: This information should be included in your methods section. You do not mention in your methods that you imaged before and after adding the mannitol solution. Also, “mannose” should be changed to “mannitol”.

Response 4: In the section 3.1, the screening of sugar related genes has been placed in “section 2.3”(Materials and Methods section). The information of “lines 162-163” has been placed in “section 2.6”. And “mannose” has been changed into “mannitol”.

Point 5: Discussion: It is not clear how the differences in transmembrane domains is important to this study. The discussion also lacks any perspective on how increased CmSWEET17 expression may affect the expression of auxin transporter genes. The author should give forth a reason for this. It seems possible that increased sucrose causes higher SWEET expression in order to store the extra sucrose. A higher presence of sucrose in the cell may then act as a signal to induce expression of auxin transport genes.

Response 5: In our study, we show that the transmembrane domains of CmSWEET17 are different from that of other species, but, at present, we don't know the relationship between the structural difference and the function of CmSWEET17. This will be one of our next research directions. In the present study, CmSWEET17 may affect the expression of auxin transporter genes, but how does it work? Whether can it affect the transportation and storage of sucrose in the axillary buds to regulate the expression of auxin transport genes? Next, we are going to buy 14C-sucrose for further study of pathway.

Point 6: Conclusion: Your conclusion lacks any mention that CmSWEET17 expression appears to contribute to the expression of auxin transport genes, which in turn leads to altered transport of auxin and ultimately altered axillary bud growth.

Response 6: In this section, we add the description of the effect of sucrose on auxin transport genes: “The data provide a preliminary confirmation of a novel function of CmSWEET17 in promoting axillary bud outgrowth. Furthermore, this function may occur via the auxin transporter pathway, increasing the expression of auxin transporters as well as auxin export from the axillary bud.”

Point 7: A few typos:

Line 186: Typo in “Phenotype”.
Line 223: Typo in “SWEEET17”
Lines 373 – 375: Should be mM, not Mm. “Scale bars indicate 1 mm” is repeated.

Response 7: We have revised the typos in lines 186, 223, 373-375.

Reviewer 2 Report

The manuscript ID: genes-652083, titled "Sugar transporter CmSWEET17 promotes bud outgrowth in Chrysanthemum morifolium" reports the upper axillary bud outgrowth that SWEET17 mediates the sugar transport in Chrysanthemum morifolium. Generally, this work is well set up and constitutively provided data to support the discussion. However, a part of your experimental settings and interpretations is lacked to be shown in your conclusion. It is my opinion that for this journal, this information, as well as more scientific interpretation of the results and more detailed methodology is required to maintain the journal's quality level. I think you need to clearly explain why this study was undertaken using C. morifolium in Introduction. It appears to be a lack of introduction.
The manuscript is worthy to be published and this journal is a journal suitable for its publication after some major revisions under reported:

1. First of all, the authors should show the characteristics of CmSWEET17 in the exogenous expression system using mammalian cells and the Xenopus oocyte. The author predicted that the function of CmSWEET17 in the phylogenetic analysis using deduced SWEET transporter A. thaliana, however, the substrates of transporters are difficult to predict and identify based on the amino acid residues.

2. The authors demonstrated that two-node stems were exposed to 20 mM sucrose and mannitol. How did the authors determine it's concentration? Similar questions have also occurred in Fig. 2. This experiment was treated at 50 mM sucrose.

3. line 79: Why did the authors collect total RNA at 10 days after exposure? I believe this outcome is not supported clearly enough.

4. The authors obtained the full-length gene of CmSWEET17 from their transcriptome library. The authors should deposit the massive parallel data using Next-generation sequencer to be available for the public to NCBI SRA. Also, which NGS did you use for your transcriptome library? Which program did you use to cut off reads? The authors never show those experimental procedures and sources.

5. Clarify the experimental setting of the particle accelerator in line 110.

6. When did you collect those samples? And, why did you collect them on that day?

7. line 224. This sentence is just result not discussion. Not to repeat. Confirm them throughout the manuscript.

8. line 251. The authors should examine other transmembrane prediction programs such as TMPRED or SOSUI. In this sentence, the author described the transmembrane regions as a conclusion. Is this the conclusion of this paper?

9. Why did the authors focus on the growing apical buds in Fig. 1, not bottom bud? Also CmSWEET9, 10 and 11 should be candidate genes.

Minor points:
1. ALL FIGURES: How many biological samples did the authors analyze? Is error bar SE or SD? Statistical analyses are lacked in all figures. Explain details in each figure legend. The authors need to show clearly for readers.

2. Fig. 4. Please show the mock control using the empty vector.

3. line 162. Mannose?

4. line 182. Change the font size of the Scientific name.

5. reference 14. PANS? Confirm them throughout the reference and arrange this journal format.

Author Response

Response to Reviewer 2 Comments

Point 1: First of all, the authors should show the characteristics of CmSWEET17 in the exogenous expression system using mammalian cells and the Xenopus oocyte. The author predicted that the function of CmSWEET17 in the phylogenetic analysis using deduced SWEET transporter A. thaliana, however, the substrates of transporters are difficult to predict and identify based on the amino acid residues.

Response 1: Thank you very much for your advice. It is a good suggestion to show the characteristics of CmSWEET17 in the exogenous expression system using mammalian cells and the Xenopus oocyte. Unfortunately, we are unable to complete this experiment at present, and we will consider this experiment in the future. Instead, we carried out transgenic engineering of chrysanthemum to verify the function of CmSWEET17. The data preliminarily confirm that the function of CmSWEET17.

Point 2: The authors demonstrated that two-node stems were exposed to 20 mM sucrose and mannitol. How did the authors determine it's concentration? Similar questions have also occurred in Fig. 2. This experiment was treated at 50 mM sucrose.

Response 2: In the chrysanthemum, 20 mM sucrose and mannitol treatments with two-node stems are based on reference [35]; and exogenous 50 mM sucrose treatment is based on reference [9].

Point 3:  line 79: Why did the authors collect total RNA at 10 days after exposure? I believe this outcome is not supported clearly enough.

Response 3: Time of collect total RNA as described previously [35]. Previous study demonstrated that, after 10 days of treatment, the phenotype of axillary bud length was change in the two-node stems treatment.

Point 4: The authors obtained the full-length gene of CmSWEET17 from their transcriptome library. The authors should deposit the massive parallel data using Next-generation sequencer to be available for the public to NCBI SRA. Also, which NGS did you use for your transcriptome library? Which program did you use to cut off reads? The authors never show those experimental procedures and sources.

Response 4: Using the existing transcriptome library (http://www.ncbi.nlm.nih.gov/bioproject/PRJNA329030) in our lab, we screened the CmSWEET17 gene. Besides, we add more information of isolation of CmSWEET17 and Sequence Analysis, including the PCR reaction and procedure and the soft of sequence analysis in “section 2.5”.

Point 5: Clarify the experimental setting of the particle accelerator in line 110.

Response 5: We have revised the description as showed in “section 2.6”.

Point 6: When did you collect those samples? And, why did you collect them on that day?

Response 6: The first sampling was around 1 p.m., so each sample was taken at the same time of day.

Point 7: line 224. This sentence is just result not discussion. Not to repeat. Confirm them throughout the manuscript.

Response 7: We have revised the all of the manuscript, and delete the duplicate statements, such as line 224 in the discussion.

Point 8: line 251. The authors should examine other transmembrane prediction programs such as TMPRED or SOSUI. In this sentence, the author described the transmembrane regions as a conclusion. Is this the conclusion of this paper?

Response 8: We have revised the conclusion, and delete the description about transmembrane regions, and add “Furthermore, this function may occur via the auxin transporter pathway, increasing the expression of auxin transporters as well as auxin export from the axillary bud”. We also use the TMPRED (https://embnet.vital-it.ch/software/TMPRED_form.html) to examine transmembrane domains (TM) of CmSWEET17 and AtSWEET17; and the result of TM also is different.

Point 9: Why did the authors focus on the growing apical buds in Fig. 1, not bottom bud? Also CmSWEET9, 10 and 11 should be candidate genes.

Response 9: After 20 mM sucrose treatment, the phenotype of upper axillary was different, and only CmSWEET17 was specifically expressed in the upper axillary bud, which is positively correlated with the axillary bud growth. The genes CmSWEET9, CmSWEET10 and CmSWEET11 were highly expressed in the bottom axillary bud, but the phenotype of bottom axillary bud was not different [35]. Therefore, we first chose CmSWEET17 for research.

Point 10:Minor points:
1). ALL FIGURES: How many biological samples did the authors analyze? Is error bar SE or SD? Statistical analyses are lacked in all figures. Explain details in each figure legend. The authors need to show clearly for readers.

2). Fig. 4. Please show the mock control using the empty vector.

3). line 162. Mannose?

4). line 182. Change the font size of the Scientific name.

5). reference 14. PANS? Confirm them throughout the reference and arrange this journal format.

Response 10: The method of Statistical Analysis is described in the “section 2.9”. We have revised the typos in lines 162,182 and reference 14.

Round 2

Reviewer 2 Report

line 131. 

It is still an unclear description of the experimental setting of the gene gun PDS-1000 such as the vacuum level, helium pressure, and others.

Author Response

Point 1: It is still an unclear description of the experimental setting of the gene gun PDS-1000 such as the vacuum level, helium pressure, and others.

Response 1: The experimental setting of the gene gun PDS-1000 has added in the “section 2.6” (marked red). Vacuum level, helium pressure and target interval were 26-28 inches Hg (mercury), 1300 psi (pounds per square inch), 6 cm, respectively.
